# Seasonal Variation in Cell Wall Composition and Carbohydrate Metabolism in the Seagrass *Posidonia oceanica* Growing at Different Depths

**DOI:** 10.3390/plants12173155

**Published:** 2023-09-01

**Authors:** Marwa Ismael, Quentin Charras, Maïté Leschevin, Damien Herfurth, Romain Roulard, Anthony Quéro, Christine Rusterucci, Jean-Marc Domon, Colette Jungas, Wilfred Vermerris, Catherine Rayon

**Affiliations:** 1UMR-INRAE 1158 Transfrontalière BioEcoAgro, BIOlogie des Plantes et Innovation (BIOPI), Université de Picardie Jules Verne, 80039 Amiens, France; marwa.ismael@u-picardie.fr (M.I.); maite.leschevin@cea.fr (M.L.); damien.herfurth@u-picardie.fr (D.H.); romain.roulard@u-picardie.fr (R.R.); anthony.quero@u-picardie.fr (A.Q.); christine.rusterucci@u-picardie.fr (C.R.); jean-marc.domon@u-picardie.fr (J.-M.D.); 2Aix-Marseille University, CEA, CNRS, BIAM, LGBP Team, 13009 Marseille, France; quent.cha@gmail.com (Q.C.); colette.jungas@univ-amu.fr (C.J.); 3Aix-Marseille University, CEA Cadarache, Zone Cité des Énergies BIAM, Bâtiment 1900, 13108 Saint-Paul-lez-Durance, France; 4Department of Microbiology & Cell Science and UF Genetics Institute, University of Florida, Gainesville, FL 32610, USA; wev@ufl.edu

**Keywords:** cell wall polysaccharides, leaves, lignin, non-structural sugars, rhizomes, summer, winter

## Abstract

*Posidonia oceanica* is a common seagrass in the Mediterranean Sea that is able to sequester large amounts of carbon. The carbon assimilated during photosynthesis can be partitioned into non-structural sugars and cell-wall polymers. In this study, we investigated the distribution of carbon in starch, soluble carbohydrates and cell-wall polymers in leaves and rhizomes of *P. oceanica*. Analyses were performed during summer and winter in meadows located south of the Frioul archipelago near Marseille, France. The leaves and rhizomes were isolated from plants collected in shallow (2 m) and deep water (26 m). Our results showed that *P. oceanica* stores more carbon as starch, sucrose and cellulose in summer and that this is more pronounced in rhizomes from deep-water plants. In winter, the reduction in photoassimilates was correlated with a lower cellulose content, compensated with a greater lignin content, except in rhizomes from deep-water plants. The syringyl-to-guaiacyl (S/G) ratio in the lignin was higher in leaves than in rhizomes and decreased in rhizomes in winter, indicating a change in the distribution or structure of the lignin. These combined data show that deep-water plants store more carbon during summer, while in winter the shallow- and deep-water plants displayed a different cell wall composition reflecting their environment.

## 1. Introduction

*Posidonia oceanica* is an endemic angiosperm seagrass living in the Mediterranean Sea between the littoral and sublittoral zones up to 40 m deep [1,2,3]. This monocotyledonous plant generates large meadows and provides important ecological functions as a habitat for sea animals [4,5] and epiphytic plants [6]. *P. oceanica* also contributes to the stability of the sea floor, removes pollutants from the sea and feeds belowground microbial communities by excreting sugars into the rhizosphere [7]. One of the remarkable properties of *P. oceanica* is its capacity to bury a significant part of its photosynthetic production, thus contributing to long-term carbon storage [8,9].

In seagrasses such as *P. oceanica* the main storage carbohydrate is sucrose, which generally represents more than 90% of the total soluble carbohydrate content [10]. *P. oceanica* contains more sucrose than starch in all tissues, and rhizomes are the main storage tissue for sucrose [11,12]. The genome of another seagrass, common eelgrass (*Zostera marina* L.), has been sequenced and possesses genes encoding proteins involved in starch and sucrose metabolism. However, there are more sucrose transporter genes and fewer starch metabolism-related genes [13].

Photoassimilates in plants are mostly incorporated into plant cell-wall polysaccharides [14]. The plant cell walls of seagrass are mostly composed of cellulose, hemicelluloses and pectins [15]. Cellulose is the most abundant cell-wall polysaccharide in seagrasses. Its content is around 20% in *Posidonia australis* but can be higher in some other seagrasses. Hemicellulose’s composition has been characterized in the *Posidonia* species from the Indian ocean, including *P. australis* and *P. sinuosa*, and is rich in xylose, indicating the presence of xylans and/or xyloglucans [16]. Their cell walls also contain galactans and arabinans, suggesting the presence of arabinogalactan proteins, as was identified in the *Z. marina* proteome [17]. Sulfated galactans have been identified in the plant cell walls of the seagrass *Ruppia maritima*, where they are mostly found in the rhizomes and roots [18]. Pectins in seagrass are enriched in an apiogalacturonan called zosterin and are weakly methyl-esterified (8–11%) [19,20]. The lower proportion of methyl-esterified pectins has been assigned to the large number (63) of putative pectin methylesterase genes annotated in the *Z. marina* genome [21]. The two Indian ocean *Posidonia* species contained more apiose in their leaves compared to the roots and rhizomes, which were instead rich in glucosyl residues [16]. *P. australis* and *P. oceanica* are rich in lignin (20–30%) [16,22,23]. The roots, rhizomes and sheaths from *P. oceanica* contain more lignin than the leaves [24]. The lignin of *P. oceanica* consists predominantly of guaiacyl (G) and syringyl (S) residues, with an S/G ratio of 0.12 in sheaths and 0.37 in rhizomes [23]. The lignin of *P. oceanica* displays a high degree of *p*-hydroxybenzoylation [22,23], whereby both G and S residues are acylated. This feature is absent in *P. australis*, indicating variation in lignin structure among *Posidonia* species [22].

Carbohydrate metabolism-related enzymes such as sucrose synthase and sucrose phosphate synthase are strongly modulated in the marine angiosperm *Z. marina* during environmental events such as elevated temperature and/or nitrate accumulation [25]. Under long exposure to low-irradiance light, *P. oceanica* reduces its non-structural sugar content by 42% in vertical leaf tissues, but not in rhizomes [26]. However, low-light environments are more favorable for *P. oceanica* growth, due to the deployment of shade adaptation strategies [27], whereas in *Z. marina* plants, the concentration of sugars and starch decreased under shading (2–3 m depth), particularly in the rhizomes [28].

The capacity for carbon storage has been shown to decline with the bathymetric distribution of the seagrass *Thalassia testudinum* [29]. In contrast, *P. oceanica* plants grow better and store more carbon in deep water [27,30].

The aim of the present study was to evaluate how the carbon assimilated during photosynthesis was (i) partitioned into non-structural and structural carbohydrates, (ii) allocated between leaves and rhizomes and (iii) the influence on two contrasted seasons and water depths on carbon storage in *P. oceanica*. Understanding the carbon distribution in *P. oceanica* under different seasons and water depths could provide new data to alleviate the influence of these parameters on carbon storage in seagrass.

## 2. Results

### 2.1. Non-Structural Carbohydrates Content

#### 2.1.1. Starch Content and Amylase Activity

The amount of starch was greater (~1.4–2.1-fold) in rhizomes than in leaves in all seasons and depths of water studied (Figure 1A). Starch was more abundant (~1.3–1.6-fold) in deep- than shallow-water plants, except in winter leaves when the starch level was similar between depths of water. In the winter, the amount of starch decreased in all tissues studied. Starch was reduced in leaves by 1.6- and 1.8-fold in shallow- and deep-water plants, respectively, when compared to summer. A 1.6-fold reduction was observed in rhizomes from shallow and deep- water plants, respectively.

As observed with starch content, amylase activity was higher (3.2–4.1-fold) in deep-water plant tissues than in shallow-water plants. (Figure 1B). The activity was greater (1.4–1.6-fold) in rhizomes than in leaves. Amylase activity did not vary between summer and winter in all tissues studied.

#### 2.1.2. Soluble Carbohydrates

The concentrations of the soluble carbohydrates sucrose, glucose and fructose were determined. In summer, the sucrose content was twice as high in rhizomes than in leaves in all water depths studied (Figure 2A), indicating a sucrose allocation efficiency. In winter, the sucrose level observed in leaves was reduced by 55% and 65% in shallow- and deep-water plants, respectively, as compared to summer plants. In rhizomes, the sucrose content was also reduced by 68% in both shallow and deep-water plants during winter as compared to summer, suggesting a reduction in sucrose translocation to sink tissues. As observed for starch and sucrose, the amount of glucose was more abundant (~1.4–1.5-fold) in rhizomes than in leaves, except in summer, when the level of glucose was not significantly different between leaves and rhizomes in shallow-water plants (Figure 2B). In winter, the glucose level decreased in leaves and that decrease was greater in leaves from shallow-water plants (31%) than from deep-water plants (16%). The level of glucose was reduced by 16% and 20% in the rhizomes from shallow- and deep-water plants, respectively, as compared to summer conditions.

As observed with sucrose, the level of fructose was 1.3–3-fold higher in rhizomes than in leaves (Figure 2C). In winter, the amount of fructose in leaves was reduced by 48% and 22% in shallow- and deep- water plants, respectively, as compared to summer. It remained unchanged in rhizomes collected from shallow-water plants, while it was reduced by 21% in deep-water plants.

These data indicate an increase of these soluble carbohydrates from leaves to rhizomes that was independent of water depth and season.

### 2.2. Composition of Cell-Wall Polysaccharides

#### 2.2.1. Non-Cellulosic Sugar Content

Since some variation in NSCs occurred under the contrasting water depths and seasonal light exposure, the cell-wall sugars content was determined (Table 1 and Appendix A). The compositional analysis of non-cellulosic polysaccharides showed that xylosyl (Xyl) and apiosyl (Api) residues were predominant in both tissues, irrespective of the water depths and seasons studied. However, some variations occurred as leaves which contained more Xyl (39–47 mol%) and Api (25–31 mol%) than rhizomes, which contained 31–33 mol% and 12–19 mol% of Xyl and Api residues, respectively. Furthermore, the arabinose (Ara) content was approximately 4–5-fold higher in rhizomes (14–16 mol%) than in leaves (3–5 mol%), except in winter for deep-water plants, when the ratio was lower (3.3-fold). Similarly, galactose (Gal) and rhamnose (Rha) levels were between 3–6- and 1.7–2.2-fold greater in rhizomes than leaves, respectively. In summer, no significant variations were observed in the non-cellulosic polysaccharide composition between shallow- and deep-water plants, except there was a slightly higher amount of apiose (13% increase) in leaves of deep-water plants. In winter, changes in the non-cellulosic components occurred in both organs and that was more pronounced in deep-water plants. Indeed, a reduction of 13% in Api residues associated with an increase of Xyl (15%) and Arabinose (36%) was observed in leaves from deep-water *P. oceanica* plants. A reduction of 21% in l-galacturonic acid (GalUA) residues likely also occurred in leaves in winter. In rhizomes, a reduction (30%) in GalUA was displayed too, while the amount of Api and Gal residues increased by 29% and 34%, respectively. Minor changes in the non-cellulosic polysaccharide content appeared in shallow-water plants in winter, albeit the level of Api residues increased by 23% and 20% in leaves and rhizomes, respectively. A similar trend was observed with GalUA residues. Additionally, the minor sugar residue GlcUA was reduced by 42% in rhizomes in shallow water, while it remained steady in other conditions. These data indicated some degradation of pectins and xylan deposition in deep-water plants during winter, while the pectin (apiogalacturonan; AGA) likely increased in shallow-water plants.

#### 2.2.2. Cellulose Content

The variation in cellulose content was similar in tissues from both deep- and shallow-water plants and both seasons (Table 1). In shallow-water plants, the amount of cellulose was 24–26% of the dry-cell-wall material (DCW) in both tissues during summer and decreased by 27% in both tissues in winter. The cellulose content was greater (~34% of DCW) in deep- than in shallow-water plants during summer. During that season, the amount of cellulose was similar between leaves and rhizomes from deep-water plants, but further decreased in winter by 50% in leaves and 35% in rhizomes. These data indicated that deep-water plants produce more cellulose in summer than shallow-water plants, but in winter, deep-water plants degrade more of their cellulose in leaves and rhizomes than shallow-water plants.

### 2.3. Lignin Composition and Enzyme Activities Involved in Lignin Biosynthesis

The concentration of lignin was generally greater in rhizomes than in leaves (Table 2). In summer, there was 1.6- and 3-times more lignin in rhizomes than in leaves from shallow- and deep-water plants, respectively. During winter, the concentration of lignin was ~17–18% higher in both leaves and rhizomes from shallow-water plants. In deep-water plants, the concentration of lignin in leaves was more than twice as high, while it was 42% lower in rhizomes.

The lignin subunit composition was analyzed using thioacidolysis, which cleaves β-O-4 linkages, the most common interunit linkage in lignin. The lignin in leaves was composed of 14–25% G residues and 75–86% S residues. The proportion of *p*-hydroxyphenyl (H-) residues was less than 3% and in most samples below the level at which the proportion could be determined with accuracy (Table 2). This composition is more similar to what is observed for hardwood lignin. The most obvious and only statistically significant difference among leaf samples from different seasons and depths was the higher proportion of S residues in the lignin from deep-water plants in the winter relative to the summer.

The lignin in rhizomes was composed of 10–16% H, 51–66% G and 22–33% S residues (Table 2). The most obvious and only statistically significant difference among the samples from different seasons and depths of rhizomes was a reduction by 30% of S residues in the lignin from deep-water samples in the summer relative to the winter.

Pyrolysis-GC-MS was used to independently validate the thioacidolysis data, especially since the proportion of S-residues was so high in the leaves. Two pyrolysis products, coniferyl alcohol (*m*/*z* 180 [M]^+^, 137; [M]^+^ denotes the molecular ion), a G residue, and 2,6-dimethoxy-4-vinylphenol (*m*/*z* 180 [M]^+^, 165), derived from an S residue, were compared in leaves and rhizomes, from shallow-water plants in the summer and deep-water plants in the winter. The ratio of the peak heights of the S- and G-derived compounds was much greater in the leaf samples than in the rhizomes samples, consistent with the results from the thioacidolysis (Appendix A).

Pyrolysis-GC-MS was also used to examine the presence of esterified phenolic compounds, specifically feruloyl and *p*-coumaroyl esters. Treatment of the samples with the methylating agent tetramethyl ammonium hydroxide (TMAH) enables the identification of pyrolysis products derived from hydroxycinnamic acids versus lignin [31] (Appendix A). The phenolic hydroxyl moieties not involved in interunit linkages become methylated. Even though the methyl ester of ferulic acid could be identified in the pyrolysate of *P. oceanica* samples (*m*/*z* 222 [M]^+^), there was no evidence of the methyl ester of *p*-coumaric acid (*m*/*z* 192 [M]^+^) (Appendix A). Instead, the methyl ester of *p*-hydroxybenzoic acid (4-methoxy benzoic acid methyl ester; *m*/*z* 166 [M]^+^) was identified. A phenol peak (*m*/*z* 94) in the native sample (i.e., not treated with TMAH) was abundant (Appendix A), although both *p*-hydroxybenzoic acid (*m*/*z* 138) and the *p*-hydroxybenzoic acid methyl ester (*m*/*z* 152) could be identified as well.

Since monolignols are derived from the aromatic amino acid l-phenylalanine (l-Phe) (although l-tyrosine may also be used to some extent [32]), its content was determined in both leaves and rhizomes (Appendix A). The amount of l-Phe was higher in leaves than in rhizomes regardless of season and depth. In leaves from deep-water plants it was more than twice as abundant as in leaves from shallow-water plants. In rhizomes, however, the amount was similar in plants from different depths in summer, but 1.4-fold greater in deep- than shallow-water plants in winter. In winter, the amount of l-Phe in leaves increased similarly (1.2–1.5-fold) at both depths. A similar trend was observed for rhizomes from deep-water plants, but the increase was greater (2-fold) in shallow-water plants. As l-Phe is a substrate for phenylalanine ammonia lyase (PAL), PAL activity was measured (Appendix A). A similar overall profile was observed between PAL enzyme activity and the amount of l-Phe in shallow-water plants, which could also be correlated to the lignin content. Unlike the distribution of l-Phe observed, PAL activity was greater (2.5–4.6-fold) in rhizomes than in leaves in summer. In winter, PAL activity increased in all tissues studied by 1.4–2.5-fold, while it decreased by 2.6-fold in rhizomes from deep-water plants. A similar trend was observed with lignin. The activity of PAL and the contents of l-Phe and lignin were correlated in shallow-water plants, but there was no obvious correlation between these parameters in deep-water plants. Another enzyme, cinnamyl alcohol dehydrogenase (CAD), which is involved the final step of the monolignol biosynthetic pathway, was assayed in leaves and rhizomes. In summer, CAD activity was similar (128 µkatal/mg of FM) in leaves and rhizomes from plants grown in shallow water as well as leaves from deep-water plants (Figure 3). The activity was greater (158 µkatal/mg of FM) in rhizomes from deep-water plants in summer, reflecting the greater amount of lignin observed in that tissue (Table 2). In winter, CAD enzyme activity increased in leaves by 30% and 64% in shallow- and deep-water plants, respectively, while it decreased by 63% in rhizomes from plants grown in deep water and remained unchanged in those of shallow-water plants. A similar pattern could be observed with lignin, except in rhizomes from shallow-water plants where the lignin content increased slightly.

### 2.4. Boron Content

Since apiogalacturonan could be involved in boron assimilation, boron content from the whole leaves and rhizomes was determined using ion chromatography (ICP-OES; Appendix A). As observed with Api, boron was more abundant in leaves than rhizomes from shallow- (1.3 times) and deep- water plants (1.6 times) in summer, but no significant difference was observed between shallow- and deep- water plants during summer. In winter, the level of boron increased in leaves and that was more pronounced in leaves from deep-water (41%) than shallow-water (15%) plants. In rhizomes, the level of boron decreased by 42% in shallow-water plants, whereas it increased (40%) in deep-water plants. No apparent correlation could be observed between the content of Api and GalUA residues and boron levels.

### 2.5. Principal Component Analysis

Principal component analysis (PCA) was performed to identify sets of variables underlying the compositional differences between samples from different seasons and water depths. Two principal components (PC) together accounted for 69% of the total variance (Appendix A). A clear separation between leaves and rhizomes was identified along PC1, which captured 57.3% of the variance (Appendix A). The PC1 loadings for Ara, Gal, Rha and GalUA sugar residues, as well as Lignin, PAL and G residues were positive, whereas those for Boron, Apiose, Xylose, S residues and l-Phe were negative. In addition, there was a weak separation between summer and winter along PC2, which captured 11.7% of the variance (Appendix A). Cellulose and glucose associated with non-structural carbohydrates contributed to PC2. The separation between shallow-water plants and deep-water plants was not well discriminated (Appendix A), probably due to the strong differences between leaves and rhizomes.

## 3. Discussion

In this study, we investigated the relationship between the metabolism of photoassimilates and the cell wall composition in *P. oceanica* leaves and rhizomes collected in contrasting depths and seasons. The main non-structural carbohydrates, starch and sucrose, were more abundant in rhizomes than in leaves and their abundance was greater in summer and in deep-water plants. Collier et al. [33] observed in different seagrass species, including *Cymodocea serrulata*, *Halophila ovalis*, *Halodule uninervis*, *Thalassia hemprichii* and *Zostera muelleri*, that rhizomes store more starch and sugars as energy reserves to maintain growth and metabolism under low-light conditions. Our results indicated that under light-limited conditions (i.e., deep water), *P. oceanica* is still able to maintain photosynthetic activity and carbon allocation in summer. Datollo et al. [27] reported that *P. oceanica* growing at a water depth of 25 m experiences a low-light environment, which reduces both light-induced damage to the photosystems and oxidative bursts. According to these authors, *P. oceanica* adapts to growing in a shaded environment by increasing the antenna size of its photosystems and this offers better growth than at 5 m depth, where light is more intense and damages photosystems. However, another study showed that *P. oceanica* displays similar amounts of the antenna proteins and the antenna size of both photosystems is similar at water depths of 2 and 26 m [34]. Overall, these authors concluded that *P. oceanica* collected at a water depth of 2 m uses more energy and activates non-photochemical quenching as a photoprotection to alleviate photodamage under high-light conditions. Procaccini et al. [35] reported that the rate of electron transport observed in *P. oceanica* was higher in shallow than deep water, suggesting that shallow-water plants, due to the greater light intensity, produce more sugars (and starch) than deep-water plants. However, another study [36] showed that the level of starch in *P. oceanica* was lower in shallow- than deep-water plants and that starch was more abundant in rhizomes than in leaves, which is consistent with our data. The lower concentration of sugars observed in shallow-water plants could be explained by the greater extent of colonization by epiphytes, which induces stress, i.e., non-photochemical quenching and ROS production, resulting in lower concentrations of starch in both leaves and rhizomes [37,38]. Based on these combined observations, *P. oceanica* collected in shallow water may actually produce more carbohydrates than deep-water plants, but since the shallow-water plants are exposed to more stress, they may use more carbohydrates to cope with that stress. Follow-up studies are needed to examine this in greater detail.

The depletion of NSC reserves observed in winter in our study suggests a negative carbon balance at both water depths. However, these reserves were still more abundant in rhizomes than in leaves, which indicates the translocation of soluble carbohydrates from leaves to rhizomes was maintained independent of the water depth and temperature (which is similar at water depths of 5 and 25 m in autumn) [27]. The reduction in the levels of all carbohydrates was more related to the light availability and quality, which differ between depths and seasons [35,39]. Under low-light conditions, the seagrass *Zostera muelleri* produced less biomass, with a reduction in the levels of soluble carbohydrates due to lower rates of photosynthesis [28,39]. Pirc [11] showed that the concentration of soluble sugars such as fructose increased from leaves to rhizomes and from summer to winter. The concentration of fructose is known to respond strongly to environmental stress, including light stress, leading to an inhibition of photosynthesis [39]. We did observe the translocation of fructose, but it was independent of depth and season. In most seagrasses, the dominant storage carbohydrate is sucrose [25,28]. Based on our study, the amount of sucrose was twice as high in rhizomes than in leaves in summer at both water depths, indicating greater photosynthesis and carbon allocation efficiencies in summer. The high proportion of soluble carbohydrates as sucrose observed in rhizomes could provide the rhizosphere with a carbon source as a nutrient for microorganisms [7]. The starch values were in the same range as those observed in a previous report [11]. The starch reserves were always more abundant in rhizomes (~2.5–6 mg of Glc/g DM) than leaves (~1.8–3.5 mg of Glc/g DM), with more starch accumulated in rhizomes from *P. oceanica* collected in deep water, regardless of the season. Govers et al. [40] showed that rhizome starch accumulation is important to overwintering seagrasses such as *Zostera noltii*, as it provides a good predictor of next year’s growth. In our study, winter plants were harvested in 2019 and summer plants in 2020. The decrease in starch observed in rhizomes in winter 2019 indicates a weak carbon demand in *P. oceanica*, which could have been advantageous to start early growth during the growing season of 2020 with more biomass, consistent with the observations for *Z. noltii* [40]. A greater amylase activity was observed in deep-water plants compared with shallow-water plants while the level of starch was mostly similar between depths, indicating that *P. oceanica* mobilizes more starch to maintain its metabolic activity in deep water, and meanwhile has the capacity to synthesize more starch than shallow-water plants. If no obvious correlation could be observed between starch content and amylase activity, PCA showed a correlation but based on organs where starch degradation and amylase enzyme activity were greater in rhizomes.

Most of the carbon fixed by plant photosynthesis is incorporated into the cell-wall polysaccharides [14]. A positive correlation between NSCs and cellulose content was observed. Similar to what was observed for NSCs, cellulose content was greater in both tissues in deep-water plants during summer and decreased in winter. The higher content of cellulose observed in deep-water plants than in shallow-water plants during summer indicated a greater availability of carbon resources in seagrass growing in deep water. That could be related to these plants experiencing less photodamage than shallow-water plants. As observed in *Halophila beccarii*, low light reduces carbon deposition in sediments to maintain carbon storage in plants [41]. The non-cellulosic sugar composition was consistent with a previous description in the literature [16], with xylose and apiose being the major non-cellulosic monosaccharides in both shallow- and deep-water plants. The high level of xylose (30–40 mol%) in tissues could arise from xylan. Peña et al. [42] showed that the xylan in duckweed (*Lemna minor*), an aquatic plant belonging to the order *Alismatales*, was mainly glucuronoxylan (GX) without any arabinose substituents. Duckweed cell walls are mostly primary walls, with a small proportion of secondary walls present in vascular tissue. The function of xylan in the primary wall needs to be elucidated [42,43]. The only significant change in xylose content was a 15% increase in leaves from deep-water plants during winter. That increase in xylosyl residues suggested an increase in xylan, which could be associated with a secondary cell wall deposition in response to winter. However, we cannot exclude a small proportion of xylose associated with xyloglucan. Pectin is a family of four galacturonic acid-rich polysaccharides including homogalacturonan, rhamnogalacturonan I (RGI), rhamnogalacturonan II (RG-II) and xylogalacturonan [44]. In some aquatic plants, including *Posidonia*, duckweed and *Z. marina*, a specific polysaccharide called apiogalacturonan (AGA) has been identified [16,19,45,46]. AGA is the major pectic polysaccharide in duckweed species and represents 48–57% of the wall mass [45]. It is less abundant (11% of the dry weight) in *Z. marina* biomass [19]. GalUA is substituted at position C-2 or C-3 with a single apiose residue or a short apiofuranose chain in AGA [15]. Galacturonic acid content decreased in deep-water plants during winter, indicating a degradation of pectin and an arrest of cell wall elongation. Unlike deep-water plants, the GalUA sugar content observed in shallow-water plants remained unchanged in leaves, suggesting at least a steady-state level of pectin. In rhizomes it increased, indicating pectin deposition. This suggests that pectin status was maintained in *P. oceanica* from shallow water in winter. Apiose is present in RGII and AGA [46]. During winter, the level of apiose, the second most abundant non-cellulosic monosaccharide present in *P. oceanica*, increased in leaves from shallow-water plants, indicating an increase in AGA, whereas it decreased in leaves collected from deep-water plants. A similar trend was observed in the rhizomes from plants collected in shallow water in winter, suggesting an increase in apiogalacturonan as well. In deep-water plants, the amount of apiose in rhizomes increased, while the amount of GalUA residues decreased, suggesting less AGA but with longer chains of apiose linked to GalUA in the cell wall of the rhizomes. The function of apiogalacturonan is not fully elucidated. In duckweed, AGA is physiologically relevant since its quantity correlates with the growth capacity of plants [47]. These authors suggested that a higher proportion of apiogalacturonan may lead to higher efficiency in the assimilation of boron as a potentially important factor to control aquatic plant growth. Boron is an essential nutrient in plants and is mainly involved in cell wall integrity but also cell membrane integrity and hormone homeostasis [48]. Our values were lower than those observed in land plants [48,49], suggesting differences in boron uptake between species. We could not observe a strong correlation between the amount of boron and the abundance of Api in relation to water depth and season, except they were both more abundant in leaves than rhizomes, which was confirmed using PCA analysis (Appendix A). However, boron was extracted from plant material but not from the cell wall, which made it difficult to determine the specific amount of boron assigned to the cell wall (Appendix A). Apiose is also known to participate in the cross-linking of two RG-II monomers by binding boron [50]. The significance of the borate cross-link between apiose residues within RGII is shown with the loss of cell–cell adhesion in the Arabidopsis *nolac-H18* (*non-organogenic callus with loosely attached cells*) mutant in which the gene *NpGUT1*, which encodes a unique glucuronyltransferase, is mutated [51]. Finally, in shallow-water plants, the increased level of Api and GalUA residues observed in winter could favor the maintenance of cell–cell adhesion, while their decrease observed in leaves from deep-water plants during winter, which was associated with an increase in Xyl residues, indicated structural changes in the cell wall of aquatic plants with lower capacity of growth.

The lignin content was higher in rhizomes and increased in winter in all tissues except in rhizomes from deep-water plants, where instead a small decrease was observed, which was consistent with the higher concentration of cellulose in that tissue. The CAD activity matched the amount of lignin, except in rhizomes from shallow-water plants. It is possible that the observed variation in CAD activity reflected the demand for other metabolites that rely on CAD, particularly in rhizomes from shallow-water plants. The observed variation in lignin concentration may instead reflect differences in the activities of peroxidases and laccases, cell-wall-localized enzymes that enable the polymerization of lignin. The lignin subunit composition differed substantially between leaves and rhizomes, with the profile of rhizomes resembling grass lignin and the profile of the leaves resembling hardwood lignin. The lignin subunit composition of rhizomes from this study was consistent with what has been reported in prior studies [22,23]. The lignin composition of the leaf tissue, performed with both thioacidolysis and Py-GC-MS, indicated a high proportion of S residues, which was unexpected. An analysis of the lignin subunit composition in *P. oceanica* by Kaal et al. [22], performed using Py-GC-MS, also indicated a higher S/G ratio in the leaf tissue than in rhizomes, but due to the semi-quantitative nature of Py-GC-MS it is not possible to determine the exact molar ratio. Pfeifer et al. [16] performed a quantitative analysis of various sea grasses using Py-GC-MS with the use of ^13^C-labeled lignin as an internal standard, but their analyses did not include *P. oceanica*, and leaf tissue was not analyzed separately for the species they compared. Pyrolysis-GC-MS data also showed the presence of feruloyl and *p*-coumaroyl esters, which tend to be abundant in grasses [31,52,53]. The data also showed the presence in rhizomes of phenol derived from both *p*-hydroxybenzoic acid and the *p*-hydroxybenzoic acid methyl ester, consistent with the reports from Kaal et al. [22] and Rencoret et al. [23].

These combined data showed that *P. oceanica* stores more carbon as NSCs, cellulose and lignin in deep water during summer and this is more pronounced in rhizomes (Figure 4). In winter, *P. oceanica* produces fewer NSCs, which is likely related to reduced photosynthesis activity, which resulted in lower cellulose content, while lignin content increased. The exception to this was rhizomes from deep-water plants, where lignin and CAD activity decreased. In addition, there was a smaller concentration of pectin and an increased concentration of xylan in deep-water plants, while the pectin concentration was higher in shallow-water plants, indicating a contrasting cell wall metabolism between shallow- and deep-water plants during winter. Further studies on monosaccharide distribution using linkage analysis would determine the fine structure of cell-wall polysaccharides specifically modulated. The results from this study are relevant in light of the rising sea levels due to global climate change, which is also expected to lead to increased turbidity. These changes may affect the growth of *P. oceanica* meadows, which in turn impacts carbon capture and storage.

## 4. Materials and Methods

### 4.1. Plant Material and Sampling Conditions

Whole individual *Posidonia* plants were harvested south of Frioul Island, Marseille, France (Coordinates 43°16′11″ N 5°17′32″ E) (Appendix A). Samples were collected by scuba divers between 9:00 and 11:00 a.m. during December 2019 and June 2020, from meadows at two different depths: 2 m (shallow water), with photosynthetically active radiation (PAR) of 600 μmol photons m^−2^ s^−1^; and 26 m (deep water), with a PAR of 40 μmol photons m^−2^ s^−1^. Plants were sampled at 3 m intervals in order to obtain representative samples from these meadows. The sampling was carried out within the framework of a protected species exemption with a prefectural decree authorizing sampling.

Immediately after harvesting, plants were kept in the dark in cold seawater. Leaves were separated from bundles and rhizomes from roots at 4 °C under green light, and epi-phyte organisms were removed through washing with seawater. The fresh plant material was then separated into leaves and rhizomes. The amount of biomass needed for the biochemical analyses required the pooling of tissue from 5–7 plants, both for leaves and rhizomes. The fresh pooled samples were ground in liquid nitrogen in a ball mill and stored frozen at −80 °C until use. All the experiments were performed on leaves and rhizomes.

### 4.2. Soluble Sugar Analyses

Glucose, fructose and sucrose contents were determined according to Duran Garzon et al. [54]. Glucose, fructose, sucrose and starch were extracted from ground frozen leaves or rhizomes (50 mg). Samples were extracted 4 times in 1 mL of 80% (*v*/*v*) ethanol at 80 °C for 20 min. After centrifugation, the supernatants were pooled, speed-vacuum dried and resuspended in water (1 mL). The soluble fraction, which corresponds to the soluble sugars, was analyzed using high performance anion exchange chromatography with pulsed amperometric detection (HPAEC/PAD) as described by De Souza et al. [55]. For quantification purposes standard curves for all measured sugars (d-Glucose, d-Fructose, d-Sucrose) were used with the following concentrations (µg mL^−1^): 50, 25, 5. Non-structural sugars were expressed in mg per g of dry mass (DM).

### 4.3. Starch Extraction and Solubilization and Amylase Activity

Starch was quantified from frozen ground powder (100 mg) after hot aqueous ethanol (80%, *v/v*, 80 °C) extraction, according to Duran Garzon et al. [54] with some modifications. Starch was pre-dissolved by adding 2 mL of 2 M KOH and stirred for 20 min at 4 °C. After 15 min of stirring, 2 mL of 1.2 M sodium acetate buffer (pH 3.8) was added to the sample. The starch was hydrolyzed into maltodextrins through the addition of 50 µL α-amylase (3000 U/mL from *Bacillus* species, Sigma-Aldrich A6814-1MU) at 75 °C for 30 min. The digestion was repeated a second time. Then, 50 µL amyloglucosidase (3300 U/mL; Megazyme; Bray, Co. Wicklow, Ireland) was added to the supernatant and incubated at 50 °C for 2 h. The digestion was repeated a second time. Finally, 100 µL of 0.8 N perchloric acid was added to stop the reaction. The resulting glucose hydrolysate was then quantified using a glucose assay (Megazyme^®^) according to the manufacturer’s instructions. Total amylase activity was determined according to the assay described by Duran Garzon et al. [54]. Frozen ground powder (100 mg) was homogenized in 2 mL of 0.1 M Na–phosphate buffer (pH 6.8). The homogenate was centrifuged at 12,000× *g* for 10 min. One milliliter of 0.1 M Na-phosphate buffer (pH 6.8), 1 mL 0.3% NaCl and 1 mL 0.5% starch (Megazyme) solution were successively added to an aliquot of 0.2 mL supernatant in a test tube and mixed thoroughly. After incubation in a water bath for 5 min at 37 °C, the reaction was stopped by adding 1 mL of 2.5 M NaOH. The hydrolyzed starch was determined using 3,5-dinitrosalicylic acid DNS method of Cui et al. [56]. The blank control was prepared as above, except that the starch solution was replaced by distilled water. Alpha-amylase activity was calculated as the difference at 520 nm between the two tubes. The standard samples were set with six levels of glucose (0, 0.5,1.0, 2.0, 3.0 and 4.0 mg/mL). Total amylase activity was expressed as total glucose produced (µg) per minute (min) and per mg of fresh material.

### 4.4. Extraction and Derivatization of Phenylalanine for GC–MS

Polar metabolites were extracted from approximately 13 mg of freeze-dried powder. The extraction protocol was adapted from Quéro et al. [57]. The sample was homogenized in 400 μL methanol and incubated using vortex mixing for 10 min at 70 °C in a thermomixer at 950 rpm. Chloroform (200 µL) was added to the vial and the extraction continued for 5 min at 70 °C and 950 rpm. After addition of water (400 μL), the mixture was vortexed and centrifuged at room temperature (10 min at 12,000 rpm). The methanol/water supernatant (50 μL) was transferred to a 2 mL tube and speed-vacuum dried for 45 min. The tubes were stored at −20 °C before derivatization and GC-MS analysis. For derivatization, the samples were speed-vacuum dried again for 30 min to remove any moisture and 40 μL of methoxyamine hydrochloride in pyridine (20 mg·mL^−1^) was added, followed by incubation for 2 h at 37 °C in a thermomixer at 950 rpm. The samples were silylated with 70 µL of *N*-trimethylsilyl-*N*-methyl trifluoroacetamide at 37 °C for 30 min under shaking at 950 rpm. Profiling of primary metabolites was conducted using gas chromatography coupled to mass spectrometry (GC-MS) analysis, following Quéro et al. [57]. The concentrations were expressed in peak area/g of dry mass (DM).

### 4.5. Cell Wall Carbohydrate Composition

#### 4.5.1. The Non-Cellulosic Monosaccharide Composition

For the non-cellulosic monosaccharide composition analysis, approximately 200 mg of frozen powder was freeze-dried and used for the cell wall extraction, as described in Duran Garzon et al. [54]. The freeze-dried cell wall material was then digested with amylase according to Foster et al. [58]. After digestion, 2 mg of destarched dry cell wall (DCW) was hydrolyzed with 2 N trifluroacetic acid (TFA) for 90 min at 121 °C. TFA was then evaporated under a stream of nitrogen. The sample was dissolved in 1 mL H_2_O and 100 µL of the sample was injected onto a CarboPac-1 column (Dionex) for high-performance anion exchange chromatography (HPAEC) of non-cellulosic sugars, which were detected using pulsed amperometry (PAD) as described in Duran Garzon et al. [54]. Analyses were performed in triplicate. For quantification purposes, standard curves for all measured sugars (l-fucose, l-rhamnose, l-arabinose, d-galactose, d-glucose, d-xylose, d-mannose, d-apiose, d-galacturonic acid and d-glucuronic acid) were used from a stock solution with the following concentrations of each sugar at 0.1, 0.3, 0.5, 0.7, 0.2, 0.8, 0.08, 0.9, 0.5 and 0.05 mM, respectively, and diluted to a half and a tenth of the concentration.

#### 4.5.2. Crystalline Cellulose Content

The cellulose content was performed as described by Foster et al. [58]. The TFA pellet was washed several times with water and freeze-dried. The sample was then incubated with 1 mL of Updegraff reagent (Acetic acid: Nitric acid: Water, 8:1:2: (*v/v/v*)) at 100 °C for 30 min. After removing the Updegraff reagent, the pellet was washed twice with water and freeze-dried. The pellet (insoluble crystalline cellulose) was hydrolyzed to glucose by adding 72% sulfuric acid (180 µL) at 30 °C for 2 h. The cellulose content of the supernatant was assayed using the colorimetric anthrone assay by determining the glucose content. The cellulose content was determined in technical triplicate for each sample.

### 4.6. Lignin Content and Subunit Composition

The acetyl bromide method was used for the total lignin extraction and quantification, according to Foster et al. [59] with modifications from Domon et al. [60]. Briefly, 100 µL of 25% acetyl bromide in acetic acid was added to each dried, destarched cell wall sample (about 1 mg). The leaves and rhizomes samples were incubated in a 50 °C water bath for 2 h and 15 min, respectively. Then, 400 µL 2 M NaOH and 70 µL 0.5 M freshly made hydroxylamine hydrochloride were added to the sample and filled with acetic acid up to 2 mL. The absorbance of the soluble acid lignin (ABSL) was read at 280 nm using a UV-Vis spectrophotometer. To determine ABSL content in *P. oceanica* the molar extinction coefficient for grasses determined by Foster et al. [59] was used.

Lignin subunit composition was determined using thioacidolysis [61], with a protocol modified after Foster et al. [59]. Approximately 5 mg dried, destarched cell wall sample was weighed into a 5 mL ReactiVial, to which 1 mL thioacidolysis reagent (87.5% (*v/v*) 1,4-dioxane; 2.5% (*v/v*) boron trifluoride etherate; 10% (*v/v*) ethanethiol) was added. The headspace was blanketed with nitrogen and the vial was incubated at 100 °C for 4 h with regular mixing. The reaction mixture was cooled and 200 μL 0.4M sodium bicarbonate was added to raise the pH. The lignin-derived products were extracted by adding 1.5 mL ddH_2_O and 1 mL ethyl acetate to each vial and vortexing for 10 s, which was followed by phase separation. The upper phase was removed with a pipette and 750 μL was transferred to a 2 mL microcentrifuge tube and dried under a gentle flow of nitrogen gas. Residual water was removed from the tube by addition of 500 μL acetone and evaporation under nitrogen gas, and this was repeated a second time. The resulting dried extract was dissolved in 500 μL ethyl acetate, 20 μL pyridine and 100 μL *N*, *O*-bis(trimethylsilyl)acetamide containing 1% trimethylchlorosilane. The samples were derivatized at 25 °C for 2 h. For rhizome samples, a volume of 100 μL was transferred to a sample vial and 100 uL acetone was added. The gas chromatography-mass spectrometry (GC-MS) analysis was performed by injecting 2 μL in the injector of a Bruker 456 GC set at 250 °C with a split ratio of 1:15 and a column flow rate of 1.1 mL helium/min. For leaf samples, 3 μL of the derivatized samples were injected without dilution in acetone, with a split ratio of 1:10. The compounds were separated on a Restek Rxi-5 ms column (30 m × 0.25 mm ID × 0.25 μm film thickness; Restek Corp., Bellefonte, PA, USA) using the following temperature gradient: 130 °C for 2 min., a 3 °C min^−1^ ramp to 270 °C followed by a 3 min hold. Lignin-derived compounds were detected using selected ion monitoring (SIM) on a Bruker Scion triple-quad mass spectrometer using electron impact (EI) ionization with 20 eV electrons, with the transfer line and source set at 250 °C. Compounds with *m*/*z* 239, 269 and 299, derived from *p*-hydroxyphenyl (H), guaiacyl (G)- and syringyl (S) residues, respectively, were monitored. Peak areas were calculated using Bruker Daltonics MS Workstation software v. 8.1.2 and used to determine the proportion of H-, G and S residues in the lignin.

Lignin subunit composition was also analyzed using pyrolysis-gas chromatography-mass spectrometry (Py-GC-MS). This method is based on the thermal depolymerization of the cell wall under anoxic conditions at 450 °C. Approximately 0.2 mg dried, destarched cell wall samples were loaded in individual quartz sample vials and analyzed based on the method of Grossman et al. [62]. The vial was placed in an aluminum sample probe and inserted into a Bruker 1079 programmable temperature vaporization (PTV) inlet of a Bruker Scion 456 gas chromatograph (Bremen, Germany). The GC was equipped with a VF-5 ms column (30 m, 0.25 mm i.d., 0.25 μm film thickness; Varian, Lake Forest, CA), which was connected to a Bruker Scion triple quadrupole mass spectrometer. The PTV injector was initially set at 100 °C and quickly ramped up to 450 °C with a helium pressure pulse of 25 psi. The split ratio was 1:50. The pyrolysate was led onto the column set at an initial temperature of 70 °C, which was maintained for 3 min. The temperature was first increased to 250 °C (4 °C min^−1^), then to 300 °C (15 °C min^−1^) for a final 2.7 min hold. The transfer line was maintained at 250 °C. Electron impact ionization was used with 20 eV electrons. The *m*/*z* range was set from 50 to 350, with a scan speed of 500 ms. Data were analyzed in Bruker Daltonics MS WorkStation software v. 8.2.1 and fragments were identified using a combination of the NIST mass spectral library and data from Ralph and Hatfield [63]. For identification of esterified phenolic acids, samples of approximately 0.2 mg inside quarts vials were soaked in 2.5% (*v/v*) tetramethyl ammonium hydroxide (TMAH; Sigma-Aldrich, St. Louis, MO, USA) in methanol [31,53,64]. After evaporation of the methanol, the sample was subjected to pyrolysis-GC-MS as described above. TMAH trans-methylates acidic hydroxyl groups during sample heating in the PTV injector.

### 4.7. PAL and CAD Enzyme Assays

#### 4.7.1. Extraction and Assay of PAL

Samples were homogenized in 100 mM Tris buffer (pH 8.8) containing 25 mM of l-phenylalanine. Phenylalanine ammonia-lyase (PAL) activity was quantified based on the production of *trans*-cinnamic acid monitored by recording the absorbance at 290 nm every 30 min for three hours at 40 °C [60]. PAL enzyme activity was expressed in microkatals mg^−1^ of fresh mass.

#### 4.7.2. Oxidation of Cinnamyl Alcohols

The enzymatic activity cinnamyl alcohol dehydrogenase (CAD) was carried out through the oxidation of coniferyl alcohol to coniferaldehyde [65]. Approximately 100 µg of extracted soluble protein fraction was used in 1 mL reaction buffer containing 100 mM Tris-HCl (pH 8.8),100 µM of coniferyl alcohol and 200 µM of NADP. The enzymatic reactions were carried out at 40 °C for 10 min and were monitored at 400 nm. The CAD activity was expressed in microkatals per mg of fresh mass.

### 4.8. Sample Preparation and Boron Analyses by ICP-OES

The samples (~50 mg of homogeneous powdered) were placed in a 300 mL beaker (BUCHI, Flawil, Switzerland). A volume of 5 mL 65% (*v*/*v*) nitric acid “Suprapur” for trace elements detection (Merck, Darmstadt, Germany) was added. The beaker was placed in a BUCHI K 425speed-digester. The samples were heated at approximately 90 °C for 4 h until the solution was translucid and the production of NO_2_ ceased. After cooling to room temperature, the solution was diluted with double distilled and deionized water up to 10 mL. A blank sample was prepared in parallel following the same method.

The samples were analyzed using Inductively Coupled Plasma Optical Emission Spectrometry (ICP-OES; PerkinElmer Avio 200, Waltham, MA, USA). The instrument was equipped with a PerkinElmer S10 autosampler. A boron standard solution for ICP (1000 ppm in 2% nitric acid) was used for calibration, following dilution to 1, 2, 5, 10, and 20 ppm). The correlation coefficient (R^2^) of the calibration curve was 0.9999.

### 4.9. Statistical Analysis

The mean values of investigated analyses in this study were compared based on the Kruskal-Wallis test [66] using R software version 4.0.3. Significant differences between the means were assessed using an α value of 0.05.

A principal component analysis was performed with mean values per parameter and conditions to study the effect of distinct conditions and the interaction between different parameters. PCA was performed in R Studio using FactoMineR package (http://factominer.free.fr/, accessed on 1 March 2023) [67].

## Figures and Tables

**Figure 1 plants-12-03155-f001:**
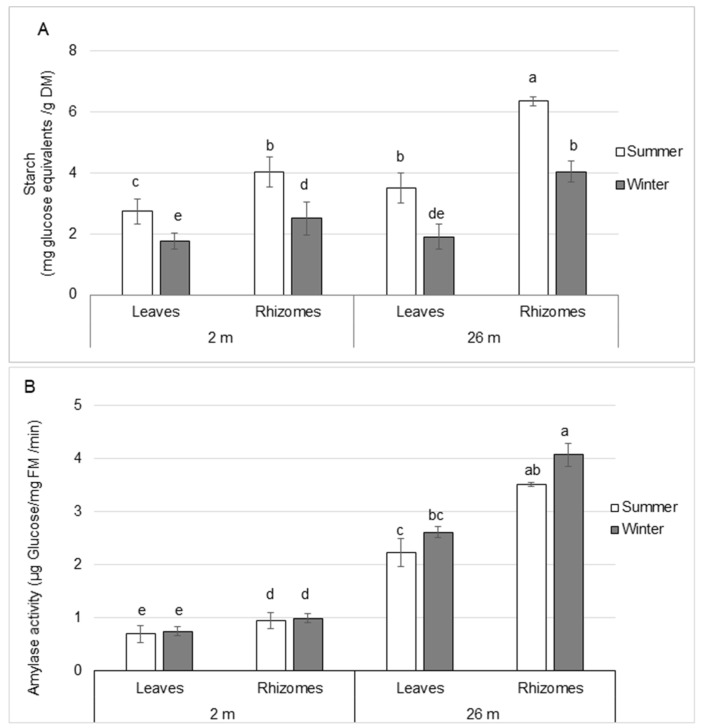
Starch content (**A**) and amylase activity (**B**) in leaves and rhizomes from *P. oceanica* harvested in summer and winter at water depths of 2 and 26 m. Data are means ± SD (n = 3). The different letters indicate significant differences (*p* ≤ 0.05) between the seasons and the depth of sampling according to Kruskal-Wallis test. DM: dry mass. FM: fresh mass.

**Figure 2 plants-12-03155-f002:**
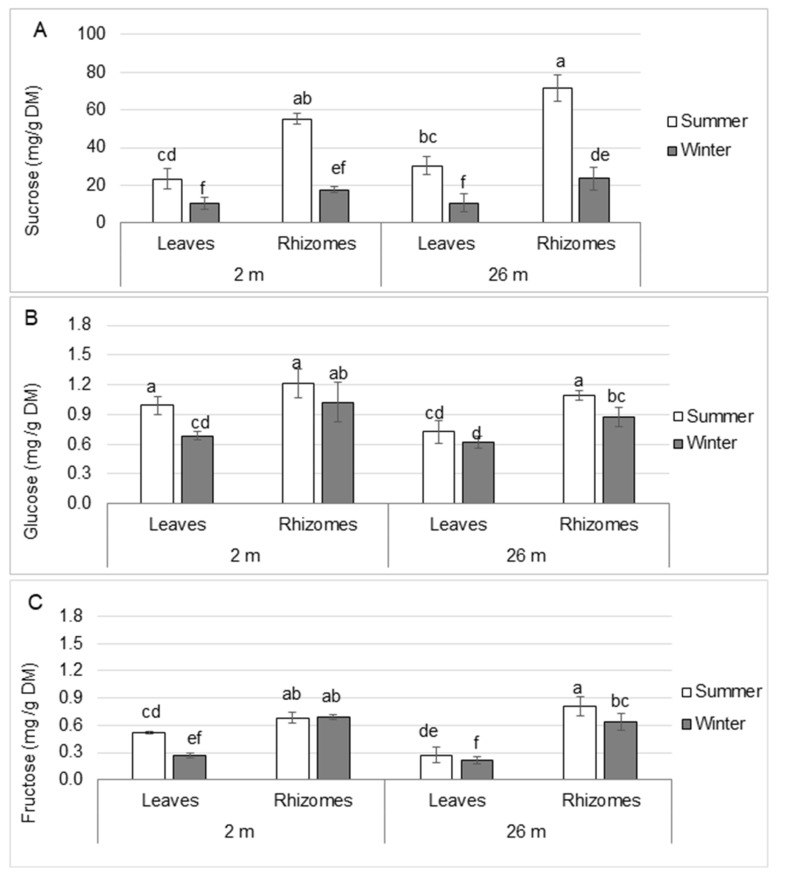
Soluble carbohydrate content in leaves and rhizomes from *P. oceanica* harvested in summer and winter at water depths of 2 and 26 m. (**A**) Sucrose, (**B**) glucose, (**C**) fructose. Data are means ± SD (n = 3). Different letters indicate statistically significant differences between samples from different seasons and/or depths of sampling (*p* ≤ 0.05) according to Kruskal-Wallis test. DM: dry mass.

**Figure 3 plants-12-03155-f003:**
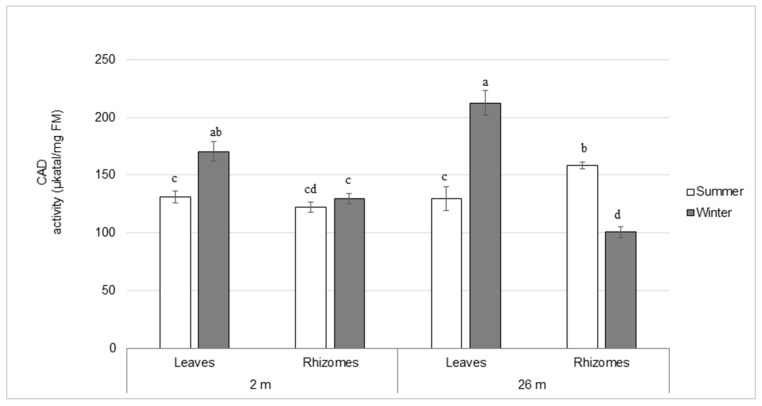
Cinnamyl alcohol dehydrogenase (CAD) enzyme activity assays in leaves and rhizomes from *P. oceanica* harvested in summer and winter at water depths of 2 and 26 m., Data are means ± SD (n = 3). The means marked with different letters indicate statistically significant differences between the seasons and the depths of sampling, (*p* ≤ 0.05) according to Kruskal-Wallis test. FM: fresh mass.

**Figure 4 plants-12-03155-f004:**
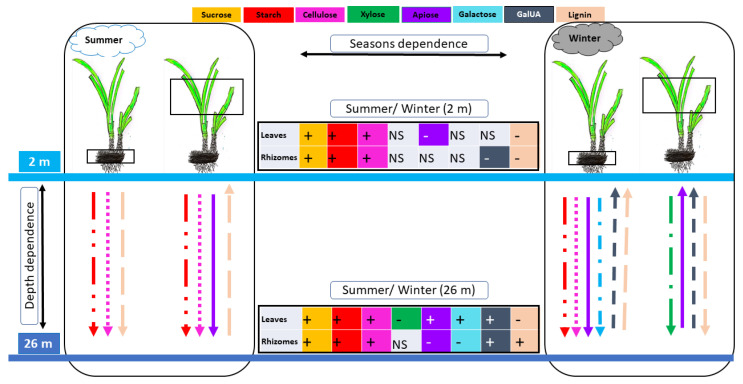
The carbon profile in *P. oceanica* meadows according to their leaves, rhizomes, the season and depth. The leaves and rhizomes studied are surrounded by a rectangle. The arrows and the tables show the significant changes in the study parameters between the two different depths and seasons, respectively. The color of the arrows and tables correspond to the study parameters as shown above. The direction of the arrows presents the changes in the parameters between the two different depths. The upward-pointing arrows mean levels are higher in shallow water. The downward-pointing arrows indicate higher levels in deep water. +: high amount in summer compared to winter, -: low amount in summer compared to winter. NS: non-significant.

**Table 1 plants-12-03155-t001:** Cell-wall sugar composition in leaves and rhizomes from *P. oceanica* grown at water depths of 2 and 26 m during summer and winter. Data are means ± SD (n = 3). Means followed by different letters are statistically significantly different between the seasons and/or the depths of sampling, according to Kruskal-Wallis test (*p* < 0.05). l-Arabinose (Ara), d-Galactose (Gal), d-Xylose (Xyl), d-Apiose (Api), d-Galacturonic acid (GalUA), d-Glucuronic acid (GlcUA). DCW: dry cell wall. Mol%: mol percentage of total non-cellulosic monosaccharides.

	2 m			
	Summer		Winter	
	Leaves	Rhizomes	Leaves	Rhizomes
*Trait*	Mean ± SD	Mean ± SD	Mean ± SD	Mean ± SD
*Cellulose (% of DCW)*	24.2 ± 3.4 *bc*	26.2 ± 3.9 *b*	17.7 ± 2.8 *de*	19.1 ± 3.2 *d*
** *Non-cellulosic monosaccharide (mol%)* **	**Mean ± SD**	**Mean ± SD**	**Mean ± SD**	**Mean ± SD**
*Ara*	3.6 ± 0.7 *cd*	16.2 ± 1.5 *a*	2.8 ± 0.3 *d*	13.8 ± 0.8 *b*
*Gal*	2.6 ± 0.4 *c*	8.2 ± 1 *b*	2.7 ± 0.3 *c*	8.1 ± 0.7 *b*
*Xyl*	43 ± 1 *ab*	32 ± 1.6 *c*	39.1 ± 1.3 *b*	32.9 ± 1 *c*
*Api*	25 ± 1.6 *b*	12.4 ± 0.7 *e*	30.7 ± 1.6 *a*	14.9 ± 1 *d*
*GalUA*	7.9 ± 1 *cd*	9.5 ± 2.4 *abc*	9.2 ± 1.1 *bc*	11.3 ± 1.3 *ab*
*GlcUA*	0.9 ± 0.2 *bc*	1.3 ± 0.2 *a*	0.8 ± 0.1 *c*	0.8 ± 0.2 *c*
	**26 m**			
	**Summer**		**Winter**	
	**Leaves**	**Rhizomes**	**Leaves**	**Rhizomes**
** *Trait* **	**Mean ± SD**	**Mean ± SD**	**Mean ± SD**	**Mean ± SD**
*Cellulose (% of DCW)*	32.4 ± 3.9 *a*	35 ± 4.2 *a*	15.7 ± 2.9 *e*	22.8 ± 2 *c*
** *Non-cellulosic monosaccharide (mol%)* **	**Mean ± SD**	**Mean ± SD**	**Mean ± SD**	**Mean ± SD**
*Ara*	3.1 ± 0.3 *d*	14.1 ± 0.8 *ab*	4.3 ± 0.2 *c*	14.3 ± 1.5 *b*
*Gal*	2.4 ± 0.5 *cd*	8 ± 0.9 *b*	1.8 ± 0.1 *d*	10.7 ± 1 *a*
*Xyl*	40.4 ± 2.1 *b*	31.5 ± 0.7 *c*	46.5 ± 1.4 *a*	32.3 ± 1.1 *c*
*Api*	28.2 ± 1.2 *a*	14.4 ± 1.2 *d*	24.5 ± 0.6 *b*	18.5 ± 1.3 *c*
*GalUA*	8.5 ± 1 *cd*	12.3 ± 1.5 *a*	6.7 ± 0.3 *d*	8.5 ± 1.2 *cd*
*GlcUA*	1.2 ± 0.1 *a*	0.8 ± 0.2 *c*	1.2 ± 0.2 *ab*	0.9 ± 0.2 *c*

**Table 2 plants-12-03155-t002:** Lignin content and subunit composition in leaves and rhizomes from *P. oceanica* harvested in summer and winter water depths of 2 and 26 m. Data are means ± SD (n = 3). The means marked with different letters indicate statistically significant differences between the seasons and/or the depths of sampling (*p* ≤ 0.05), based on Kruskal-Wallis test. DCW: dry cell wall. *p*-hydroxyphenyl (H), guaiacyl (G) and syringyl (S) residues.

	2 m			
	Summer		Winter	
	Leaves	Rhizomes	Leaves	Rhizomes
*Trait*	Mean ± SD	Mean ± SD	Mean ± SD	Mean ± SD
*Lignin (% of DCW)*	8.8 ± 0.8 *f*	13.8 ± 0.7 *d*	10.3 ± 0.9 *e*	16.4 ± 0.7 *b*
** *Phenylpropanoid units (%)* **	**Mean ± SD**	**Mean ± SD**	**Mean ± SD**	**Mean ± SD**
*H*	tr *	10.4 ± 1.1 *bc*	tr *	12.4 ± 0.5 *b*
*G*	24.7 ± 4.4 *d*	62.9 ± 2.4 *ab*	21.9 ± 4.2 *d*	65.6 ± 0.8 *a*
*S*	74.7 ± 3.8 *b*	26.7 ± 1.6 *c*	78 ± 4.2 *ab*	22 ± 1.3 *d*
*S/G*	3.1 ± 0.7 *b*	0.4 ± 0.04 *cd*	3.7 ± 0.8 *b*	0.3 ± 0.02 *e*
	**26 m**			
	**Summer**		**Winter**	
	**Leaves**	**Rhizomes**	**Leaves**	**Rhizomes**
** *Trait* **	**Mean ± SD**	**Mean ± SD**	**Mean ± SD**	**Mean ± SD**
*Lignin (% of DCW)*	7 ± 0.4 *g*	20.9 ± 0.8 *a*	15.4 ± 1.3 *c*	12 ± 0.8 *d*
** *Phenylpropanoid units (%)* **	**Mean ± SD**	**Mean ± SD**	**Mean ± SD**	**Mean ± SD**
*H*	tr *	15.5 ± 3 *a*	tr *	15.4 ± 2.1 *a*
*G*	22.8 ± 4.3 *d*	51.3 ± 0.4 *c*	13.5 ± 2.3 *e*	61.3 ± 3.7 *bc*
*S*	76.3 ± 2.7 *b*	33.2 ± 0.6 *c*	85.5 ± 1.2 *a*	23.3 ± 2.3 *d*
*S/G*	3.5 ± 0.9 *b*	0.6 ± 0.02 *c*	6.5 ± 1.2 *a*	0.4 ± 0.1 *de*

tr * = trace; signal too low for accurate quantification; it was not possible to calculate SD.

## Data Availability

Not applicable.

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
