# Peer review of "Seasonal Variation in Cell Wall Composition and Carbohydrate Metabolism in the Seagrass Posidonia oceanica Growing at Different Depths"

_plants, 2023, doi:10.3390/plants12173155_

Round 1

Reviewer 1 Report

In the Manuscript "Seasonal variation in cell wall composition and carbohydrate metabolism in the seagrass Posidonia oceanica growing at different depths" by Ismael et al., the authors collected data on carbon distribution under 2 different depths and 2 seasons in a seagrass species. The introduction tries to get into the topic, but it is partly very bumpy and provided with facts that are not absolutely necessary for the work. The results section reads very heavily and should be revised. There are also some major errors and some inconsistencies (see comments below). The discussion tries to put the results into context. Unfortunately, this is not always successful and is difficult to read. References are sometimes incorrect and repetitions of results unnecessarily inflate the text. The situation is similar with the material and methods section. In part, methods are insufficiently (not comprehensibly) referenced (soluble sugars) and on the other hand, other less important methods (boron determination) are described in great detail. In conclusion, it can be said that the work contains a good collection of data, which, neatly processed and discussed with current literature, can make a good contribution to knowledge about seagrasses. However, the manuscript needs to be fundamentally revised. I would therefore reject it with "major revisions" and propose it for resubmission.

General concerns:

According to the measurement data presented, the cell wall fraction consists of 15-34 % cellulose and 7-20 % lignin. The monosaccharides are only given in relative terms. What do the remaining 46-78 % of the cell wall consist of? The aim of the work is to describe carbon distribution in the cell wall. However, a large part is not explained at all. Thus, the title and the objective promised too much?

 Starch, soluble sugars, amylase activity and phenylalanines were determined in whole plant organ material. Single sugar content of the cell wall, cellulose content, lignin in extracted cell walls. For PAL and CAD enzyme assays and boron determination, it remains unclear which fraction was used (assuming whole plant organs). It is essential that this is stated precisely through the whole manuscript.

 Special concerns:

50-53     redundant (does not play any role in the subsequent manuscript)

62           also Xylans, Mannans, Xycloglucans, Pectins ect.

97 ff       The aim of the study should be made clearer. The effects can actually only be attributed to the water depth and the season. Values about temperature as well as light were not given at all and can therefore not explain any effects

100 ff    Anticipating the results at this point I think is inconvenient and does not make sense here.

109-111 This is redundant at this point and should be included in the next section if at all.

114 ff    The starch content ranges from 0.02 to 0.098 % of the dry matter. That seems very low to me (see also Fig 1).

122         Due to the insufficient presentation of the amylase activity determination in the material and methods, it cannot be clearly understood here what amylase activity means. Conversion to sugars, actual enzyme activity, or something else? See also Figure 1b: Glucose/Protein and time. How much protein is in the tissue? What is the relation between the glucose produced and the total concentration in the tissue? Why is amylase activity no longer mentioned in the discussion?

129-130 It can be assumed that the amylase activity is determined more by the physiological state than by the amount of starch available. To conclude that there is a negative correlation is therefore not causal. Moreover, such a statement should rather be given place in the discussion.

177 ff    The specification in mol% cannot be reconstructed. What is meant here? % of total sugar in the cell wall? Why do the values not add up to 100 % (Table 1)? Why is the absolute mass not given? Were standard curves determined for the HPAEC?

198 ff    This section on boron is out of place here, since according to the heading it is the sugars that are to be discussed here.

237         please specify G-residues, S-residues and H-residues

265-281 Is rather discussion and should not be in the results section as it distracts from the actual results.

292         The same applies here as for amylase (and CAD line 295 ff). The activity of an enzyme does not correlate with the product. Rather, the product is dependent on the substrate and the workload of the enzyme. This should then also be formulated in this way.

350         Govers et al: Reference is not correct. In the manuscript given, neither starch nor its distribution is discussed.

361         The authors claim that biomass production is greater at depth than in shallow water. They justify this with photodamage but do not evaluate the different light intensities/energy availability for photosynthesis. Such statements require clear data, which is not available here (light measurement, photosynthesis measurement).

433         In this section, the greater storage of carbon at depth is explained by the better water quality. The evidence for this is also lacking (see comment on 361 above).

411-417 redundant (result section)

465         It is not clear whether the samples were pulverised in the dried state and if so, how they were dried.

470         Without further literature, it is unfortunately not possible to understand how the sugar analysis and the determination of the amylase activity were carried out.

500         Peak area/g DW is not permitted. An internal standard should be used.

504         It is not clear what you mean by “frozen powder” please specify and explain how it was produced (see above remark to 465)

507         reference format should be revised

509         100 µl (of the sample?) was injected

515         Foster et al. [48] is the wrong reference, there is a part II of this work that should be cited instead

517         “Updegraff” please specify

586-587 redundant, please include in the section afterwards

In general: It is imperative that authors revise punctuation throughout the manuscript. Subscript characters should be used (H2O, NO2 ect). Unit labelling should be standardised (space between number and unit).

Reviewer 2 Report

Dear author,

this is an interesting, well-written and important paper, as knowledge on seagrass cell walls is still strongly limited and necessary to protect these important ecosystems.

I have some minor comments:

Materials and methods:

4.2. Please explain the methods shortly; I have no possibility to read the literature 46

Results:

Table 1. I´m not sure whether an accuracy of two digits after the decimal point is appropriate; I would prefer one digit after the point comparable to Table 2.

Discussion;

Figure 4: The Figure combines all results but is difficult to understand at the first view. I would change at least the colours for starch and lignin, as it is a little bit difficult to distinguish between the both red colours.

Reviewer 3 Report

Dear authors, dear editors,

I hereby provide a review on the manuscript “Seasonal variation in cell wall composition and carbohydrate metabolism in the seagrass Posidonia oceanica growing at different depths” by Ismael et al.

The manuscript deals with variations in carbon storage of this marine species with a focus on the cell wall components lignin, starch, sucrose and cellulose. It provides information about carbon fluxes during seasons and water depths and therefore adds very well to the increasing interest in “blue carbon”.

Overall, the quality of the manuscript is high and I really appreciated reading it. I am confident that the manuscript is publishable after a few very minor revisions. Please find below some short comments:

1.       Figure 4 is not easy to understand. Especially, the increase and decrease of contents are not intuitively understood by using flat bars. Maybe it would be helpful to use triangles with the flat side to show the increase and the pointed side to show decrease. Furthermore, the colors are very similar and not easy to differentiate. I propose to separate the depth and the seasonal variation into two separate panels of this figure. This would of course lead to a less condensed form and give some information two times.

2.       I wondered whether the authors have performed a principal component analysis (PCA) of the different samples. Even though the sample size is relatively low it could help to see distinct clusters in the dataset. Evaluate of the loading plots would enable to see correlating trends in the sugars. You mentioned “no apparent correlation could be observed between the contents of Api and GalUA and boron ion levels” (ll. 206+207). Maybe, there are some non-apparent correlations becoming visible with this analysis.

3.       I really appreciated reading your discussion as it gives a detailed insight into the existing data and where your study adds to this. You referenced Atmodjo et al. (2013) which is a good review on pectin components in general. Unfortunately, there is only a very short part on apiogalacturonan. There is a recently published review article only on that pectic macromolecule (https://doi.org/10.1002/9781119312994.apr0803). Maybe, this could further add to your discussion.

4.       You wrote “Additionally, boron was extracted from plant material but not from the cell wall […]” (ll. 399+400). Was there any specific reason for that? Would it not be possible to extract a so-called alcohol-insoluble residue, which is common in the cell wall community, and test this for boron contents?

As I conclusion, I can only repeat my above-mentioned statement that I recommend minor revisions of the manuscript. The performed research is well conducted and fits nicely to the journal.

Round 2

Reviewer 1 Report

The authors have partly tried to respond to the comments. To a certain extent, they have also succeeded in making the manuscript more comprehensible. However, some substantial questions are still open. In particular, the discussion and the conclusions contained therein are partly very vague and not sufficiently substantiated. Either measurement data (photosynthesis rates, growth rates, actual light conditions, etc.) or a sufficiently logical and comprehensible discussion based on literature is missing. I would return the manuscript for revision. However, in a revised state it may provide valuable data for subsequent work.

Concern 1:

2.1.1

Even if you found a calculation error that corrected the starch contents in your samples, I still think that your starch content is far too low. If compared with a quick literature search, starch content in P.o. ranges from 15 up to >100 mg/g DW. Is there any explanation why you content is so low (between 1.8 -6.5 mg/gDW)? (see for example ref 36)

Concern 2:

2.2.1

It has now been better explained what mol% means. However, it is still not understandable why no absolute values (or %of DCW) of the soluble carbohydrates are given. This would enable the comparability/relationship to all other carbohydrate pools.

In addition, it would be very helpful if the mass ratio of cell wall to total biomass were also given.

Concern 3:

Figure S4

Boron content of 2.5 g /kg seems really high. (normally in plant tissue it is about 5-60 mg/kg). Please check if the numbers are correct and give an explanation for three high contents

Concern 4:

410-414 You can’t conclude from higher photoprotection in shallow water a higher carbohydrate production in the depth if you do not know absolute photosynthetic rates AND the available photosynthetically available radiation! See comment in the first review.

Your previous answer:” , we maintain our statment that P. oceanica collected at 2 m depth of water uses more energy and activates non-photochemical quenching as a photoprotection to alleviate photodamage under high light.“ I do not doubt that, but it does not explain the differences in carbon distribution. E.g. if due to lower available energy in the depth also growth rate is decreased, you will observe different carbon contents/distribution which are due to different growth but not to higher photoprotection at 2m

Concern 5:

L432 ff

“Starch reserves were always more abundant in rhizomes (~4-6 mg of Glc/g DM) than leaves (~3-4 mg of Glc/g DM), with more starch accumulated in rhizomes from P. oceanica collected in deep water, regardless of the season.”

Values are not the same as in Fig 1. Please correct! (see also comment to 2.1.1)

Concern 6:

L448

“Most of the carbon fixed by plant photosynthesis is incorporated into the cell wall polysaccharides [13].”

I cannot understand this statement on the basis of the reference. The authors of the reference only describe that up to 90 % of the C is stored as sucrose in the roots of Zostera.

Concern 7:

L451

The higher content of cellulose observed in deep-water plants than in shallow-water plants during summer indicates a higher biomass production in seagrass growing in deep water.”

I also consider this conclusion to be unsubstantiated.

L490-495: See comment to Fig S4. Your B – contents are implausible and should also be discussed here

Concern 8:

L494 “Figure S1 5” please correct

Concern 9:

L529 ff previous concern:

> 433         In this section, the greater storage of carbon at depth is explained by the better

> water quality. The evidence for this is also lacking (see comment on 361 above).

> Based on the cited literature in the manuscript (ref 27 and 33, page 24), we consider the

> better water quality the most likely cause of the higher carbon storage observed in plants

> growing at greater depths.

This objection was not considered. The authors only refer to earlier work in their responses. However, the text was not supplemented. It is thus still not comprehensible why and whether better water quality can serve as an explanation.
